# Phospholipid Membrane Transport and Associated Diseases

**DOI:** 10.3390/biomedicines10051201

**Published:** 2022-05-23

**Authors:** Raúl Ventura, Inma Martínez-Ruiz, María Isabel Hernández-Alvarez

**Affiliations:** 1Departament de Bioquímica i Biomedicina Molecular, Facultat de Biologia, Universitat de Barcelona, 08028 Barcelona, Spain; raulventura99@hotmail.com (R.V.); maculadar@hotmail.es (I.M.-R.); 2Centro de Investigación Biomédica en Red de Diabetes y Enfermedades Metabólicas Asociadas (CIBERDEM), Instituto de Salud Carlos III, 28029 Madrid, Spain; 3IBUB Universitat de Barcelona—Institut de Biomedicina de la Universitat de Barcelona, 08028 Barcelona, Spain

**Keywords:** glycerophospholipid, Mfn2, oxidized phospholipid, lipid transport proteins, membrane contact sites

## Abstract

Phospholipids are the basic structure block of eukaryotic membranes, in both the outer and inner membranes, which delimit cell organelles. Phospholipids can also be damaged by oxidative stress produced by mitochondria, for instance, becoming oxidized phospholipids. These damaged phospholipids have been related to prevalent diseases such as atherosclerosis or non-alcoholic steatohepatitis (NASH) because they alter gene expression and induce cellular stress and apoptosis. One of the main sites of phospholipid synthesis is the endoplasmic reticulum (ER). ER association with other organelles through membrane contact sites (MCS) provides a close apposition for lipid transport. Additionally, an important advance in this small cytosolic gap are lipid transfer proteins (LTPs), which accelerate and modulate the distribution of phospholipids in other organelles. In this regard, LTPs can be established as an essential point within phospholipid circulation, as relevant data show impaired phospholipid transport when LTPs are defected. This review will focus on phospholipid function, metabolism, non-vesicular transport, and associated diseases.

## 1. Introduction

Phospholipids are one of the main components in eukaryotic membranes, with glycerophospholipids (phosphatidylcholine, phosphatidylethanolamine, and phosphatidylserine) being particularly significant.

In order to develop their function, phospholipids are transported between organelle structures through vesicular or non-vesicular transport mediated by lipid transport proteins (LTPs). Recent studies indicate that phospholipids are responsible for several diseases related to either their deficient transport or oxidized phospholipids, increasing their importance in recent years.

Specifically, oxidized phospholipids are one of the most pathogenic elements related to phospholipids and they have been associated with a plethora of diseases, such as atherosclerosis or non-alcoholic steatohepatitis (NASH).

The goal of this review is to assess the potential of phospholipids to function as key elements in metabolism. We included aspects of phospholipid synthesis and their transport across the organelle contact sites. In addition, we included the specific defects in transport mechanisms involved in diseases.

## 2. Phospholipids Structure, Synthesis, and Classification

### 2.1. Glycerophospholipid

Glycerophospholipids are the most abundant phospholipids in mammalian cell membranes. They have a common structure formed by a polar head (phosphate group), and two fatty acids attached through esters bonds, to the first and second carbons of a glycerol molecule. These fatty acids determine every phospholipid structure, type, and function, and endow an amphipathic character due to its hydrophobicity, which is added to hydrophilicity contributed by the phosphate group [1]. However, phospholipids can have more bonded elements, such as serine or ethanolamine, also determining their typology.

As a result of this amphipathic character, phospholipids can form bilayer structures present in lipid membranes, for instance, in the plasma membrane. Particularly, hydrophobic groups are orientated inward, leaving the polar head in contact with an aqueous phase. This arrangement allows cellular organelle delimitation and anchor of many important molecules, among others [1].

Phosphatidylserine (PS), which has a negative charge and a crucial role in apoptosis, is one of the most relevant glycerophospholipids. It is constituted of two fatty acids and a phosphodiester bond with serine attached to the third carbon, forming glycerol [2]. On the other hand, both phosphatidylethanolamine (PE) and phosphatidylcholine (PC) have a phosphate group combined with ethanolamine and choline, respectively. This complex is also bonded with glycerol through its third carbon [3,4].

On the other hand, phosphatidylinositol (PI) consists of two fatty acids attached to a phosphate group-bonded glycerol, which is also bonded to an inositol group. It approximately represents 10–15% of total lipids, being especially important in Golgi apparatus, endoplasmic reticulum (ER), and the outer mitochondrial membrane [5].

However, phosphatidic acid (PA) is the less predominant lipid (1–2% of total cell lipids). It is constituted of a glycerol molecule with two fatty acids esterified to three hydroxyl groups and a phosphate group. Usually, one fatty acid is saturated, while the other is unsaturated [6].

Finally, another relevant phospholipid is cardiolipin (CL), a diphosphatidylglycerol lipid. Although it is found only in mitochondria, mostly in their inner membrane, it only represents 2–5% of total lipids. CL has a characteristic structure, consisting of two phosphate residues and four fatty acid chains (it is constituted by two PA moieties connected to a glycerol, forming a dimeric molecule). CL is formed by four alkyl groups, giving the structure two negative charges [7,8].

Since membranes are not homogenous, every membrane and subdomain have a specific phospholipid composition. Nevertheless, phosphatidylcholine is the most abundant glycerophospholipid, making up approximately 40–50% of total phospholipids [1].

### 2.2. Phospholipids Synthesis

Mammalian cells synthesize phospholipids using a precursor, diacylglycerol (DAG), which derive from PA. In general, a great variety of enzymes located in the ER, mitochondria, and cytosol are needed to synthesize phospholipids (Figure 1). First, to generate the aforementioned phosphatidic acid from glycerol-3-P, two ER-and-outer-mitochondrial-membrane acetyltransferases (glycerol-3-phosphate acyltransferase, GPAT, and 1-acylglycerol-3-phosphate-*O*-acyltransferase, AGPAT) act sequentially. Once it is synthesized, the phosphatidic acid phosphatase (PAP), catalyzes DAG formation. Specifically, PAP is only activated when it contacts to the ER membrane. From DAG, phosphatidylcholine, phosphatidylserine, and phosphatidylethanolamine will be synthesized [9,10,11]. Generally, phosphatidylcholine is synthesized via the Kennedy pathway in mammals. Essentially, choline is imported into the cytoplasm, where it is phosphorylated, obtaining phosphocholine. This modification is done by choline kinase (CK). Afterwards, choline-phosphate cytidylyltransferase (CCT) generates cytidine-5-diphosphocholine (CDP-choline).

Finally, CDP-choline: 1,2-diacylglycerol cholinephosphotransferase, (CTP) catalyzes the transfer of DAG to CDP-choline, generating the PC within the ER membrane. Alternatively, phosphatidylethanolamine can be converted again to PC through three methylations catalyzed by PE *N*-methyltransferase (PEMT) in the ER membrane. Despite it not being a predominant pathway, hepatocytes are the only cell type in which this reaction produces significant amounts of PC [12,13].

In the case of phosphatidylserine, two enzymes can be used in its synthesis: phosphatidylserine synthase 1 (PSS1) and phosphatidylserine synthase 2 (PSS2), which are in a specific area formed by mitochondrial and ER membranes called mitochondrial associated membranes (MAMs). Specifically, PSS1 exchanges the choline of a PC for a serine, generating a PS molecule. However, PSS2 changes the ethanolamine of PE for a serine, also generating PS [11,14,15].

To synthesize PE, an inner mitochondrial membrane-located enzyme decarboxylates PS, generating PE. However, there is also another pathway like PC synthesis, whereby ethanolamine is phosphorylated by ethanolamine kinase and then converted to CDP-ethanolamine. CDP-ethanolamine is finally converted to PE when DAG is incorporated [14,15]. Although PC, PE, PS, PI, CL, and PG are all derived from phosphatidic acid, they follow different routes.

In the case of PI synthesis, it exclusively occurs in the endoplasmic reticulum. This process needs the sequential action of two enzymes, cytidine diphosphate diacylglycerol synthetases 1/2 (CDS1/2), and phosphatidylinositol synthase (PIS). CDS1/2 catalyzes phosphatidic acid, obtaining cytidine diphosphate diacylglycerol (CDP-DAG), which is a PIS substrate. PIS is a transmembrane protein able to perform a reversible reaction, producing PI (this conversion occurs very occasionally and under specific circumstances) [16,17,18].

However, CL and phosphatidylglycerol (PG) are synthesized in the mitochondria. Firstly, PA is converted to CDP-DAG by the mitochondrial translocator assembly and maintenance protein 41 (TAMM41). CDP-DAG is then transformed to phosphatidylglycerophosphate (PGP), and finally the protein tyrosine phosphatase mitochondrial 1 (PTPMT1) catalyzes its conversion into PG. PG is the substrate for cardiolipin synthase (CLS), obtaining CL. Once the CL is synthesized, it is remodeled to achieve its final conformation with predominantly unsaturated fatty acids. This process occurs through the sequential action of several enzymes, such as cardiolipin-specific deacylases and transacylases (Tafazzin) [16,19,20].

## 3. Transport of Phospholipids

The ER is a crucial site of phospholipid synthesis. From this organelle, large molecules of phospholipids are delivered to other cellular membranes in order to keep their lipid compositions.

Phospholipid transport can be classified as non-vesicular or vesicular. Mounting evidence propose MCS (membrane contact sites) as specific transient or stably cytosolic gaps between two organelle structures where the transport is faster [21,22].

Additionally, phospholipid transport in this area is accelerated due to lipid transfer proteins (LTPs). LTPs are lipid carriers that bind monomeric lipids creating a hydrophobic space, transporting them between two membrane structures through an aqueous phase [23]. These specific transfer proteins can modify the lipid distribution of membranes, leading to changes in properties such as curvature or fluidity. The main feature of LTPs are their domains [OxySterol Binding Protein (OSBP)-Related Domain (ORD), synaptotagmin-like mitochondrial-lipid binding protein (SMP), Acyl-CoA, phosphatidylinositol transfer protein (PITP)] and the number of domains they contain, which can be one or more (single-domain or multi-domain proteins, respectively). Based on their lipid-binding capacity, LTPs can be differentiated into three groups: (1) sphingolipid, (2) sterol, and (3) phospholipid-transfer proteins [24,25].

### 3.1. Phospholipid Transport between the ER-Plasma Membrane

The plasma membrane (PM) is particularly enriched in phosphoinositides and phosphatidylserine. Phosphoinositides are generated by the phosphorylation of PI in the ER, which is then transported through PITP [26]. Afterwards, PI is phosphorylated to phosphatidylinositol 4-phosphate (PI4P), which can be further phosphorylated, generating phosphatidylinositol 4,5-biphosphate (PI(4,5)P_2_) within the plasma membrane. They function as precursors of second messengers. Receptor-regulated phospholipase C (PLC) hydrolases PI(4,5)P_2_ generating inositol 1,4,5-triphosphate (IP_3_) and DAG. Since PI(4,5)P_2_ levels can drop easily during PLC signaling (Figure 2a), the transfer of lipid intermediates between the ER and PM is highly required to ensure stable levels of this key lipid [27].

The produced DAG is converted into PA, being transported back to the plasma membrane from the ER. CDP-DAG synthesis from the transported PA by CDS enzyme is another key step in PLC signaling. Finally, PI is synthesized from CDP-DAG, which can be transferred to the ER [28].

PITP proteins were primarily identified as soluble factors. The first in vivo evidence of ER-PM PI transport was found in drosophila photoreceptors. The microvillar PM and ER-derived sub-microvillar cisternae (SMC) create a structure similar to MCS, where retinal degeneration B (RdgBα) is localized and transports PI through its PITP N-terminal domain.

In the case of mammals, studies with cell lines overexpressing membrane-associated phosphatidylinositol transfer protein 1/2 (PITPNM1/2 also calledNir2/Nir3) receptors have been performed [29,30]. However, there is still not enough evidence that supports their function in PLC signaling, and as these studies have been done with overexpression of the receptor, it is suggested that these proteins might only be required in acute stimulations.

Another class of LTPs are the extended synaptotagmins (E-Syts). They are localized as integral membrane proteins in the ER. Their structure contains several antiparallel β-sandwich that contains three Ca^2+^-binding loops (C2 domains) and an SMP (synaptotagmin-like mitochondrial lipid-binding) domain, which has no selectivity for any glycerolipid. This SMP domain dimerizes, creating a cylinder where two PL can fit. In mammals, E-Syt1, E-Syt2, and E-Syt3 are found. The E-Syt family is only active when Ca^2+^ levels are elevated after PLC activation. Calcium binds to C2 domains, promoting E-Syt2/3 tethering with PM through PI(4,5)P_2_, whereas E-Syt1 translocates to ER-PM MCS [31,32].

The Phospholipid Transfer Protein C2CD2L (also known as TEM24 or C2CD2L) is a protein greatly enriched in pancreatic β-cells, playing an essential function in glucose-sensitive insulin release regulation. It is an SMP domain protein anchored at the ER. Its SMP domain only binds one molecule of PI, and it concentrates at ER-PM MCS during resting conditions. When phosphorylated by protein kinase C, TEM24 dissociates from PM, ceasing PI transport, while when dephosphorylated by phosphatase PP2B, it associates again [33].

Regarding PS transport (Figure 2b), there are other types of proteins, such as Oxysterol-binding protein (OSBP)-related protein 5 or 8 (ORP5 and ORP8), which are in charge of PI(4,5)P_2_ and PI4P removal from the PM while transferring PS from the PM to ER. In order to create a gradient for this exchange, phosphatidylinositol-3-phosphatase SAC1 decreases PI4P levels at ER-PM MCS [34].

### 3.2. Phospholipid Transport between the ER-Mitochondria

Mitochondria are known to be the power house of the cell. They are responsible for energy obtention derived from the breakdown of carbohydrates and fatty acids, which are transformed into adenosine triphosphate (ATP) in a process called oxidative phosphorylation. However, the functions that this energy-producing organelle can develop go beyond this.

Mitochondria can produce most phospholipids with the exception of PC, PI, and PS, which have to be transported from the ER. Inside the mitochondria, PE is synthesized from PS and then imported back to the ER, where it can be converted into PC [35].

Since phospholipid exchange between ER-mitochondria through an aqueous space is a very low process, proximity between these two organelles is needed to promote phospholipid transport. In this regard, MAMs function as a bridge connecting both organelles and this formation is associated with calcium flux and phospholipid transport (Figure 3) [36]. This close localization provides a hydrophobic shortcut for phospholipid non-vesicular transport, allowing avoidance of the aqueous area.

In this specific area, ORP5/8 has been identified in mammalian cells, where it transports PS from the ER to mitochondria [37].

On the other hand, it is relevant to mention that this transport could also be facilitated by a mitochondrial protein called Mitofusin 2 (Mfn2). Mitofusin 2 and its homolog, Mitofusin 1 (Mfn1), are transmembrane nucleotide guanosine triphosphate phosphatase (GTPases) located on the outer mitochondrial membrane. Particularly, Mitofusins play an essential role in the mitochondrial fusion process in addition to Optic Atrophy Protein 1 (OPA1), an inner mitochondrial membrane protein [38].

Despite their major relevance in maintaining mitochondria morphology and integrity, Mitofusins have extra functions associated with phospholipid transport. It has been demonstrated that Mfn2 is enriched in MAMs where the tethering of ER mitochondria occurs by physical interaction of Mfn1 or 2 in the outer mitochondrial membrane. Recent data show that Mfn2 ablation or reduction in the liver causes a disruption in phosphatidylserine transport from the ER to the mitochondria [39]. Mfn2 binds and creates rigid regions enriched in PS close to MAMs, facilitating its transport by proteins such as ORP5 or ORP8 from the ER to the mitochondria. In vivo, L-serine incorporation experiments were developed to monitor Mfn2 capacity to bind and transfer PS [39]. This investigation showed that Mfn2 liver knock-out promoted less L-serine incorporation into PS, PE, and PC, leading to endoplasmic reticulum stress and the progression of a NASH-like phenotype and liver cancer, which will be discussed further.

In regard to Mfn1, the same experiments were developed as with its homolog to test its phospholipid-binding capacity. In fact, the data demonstrated that contrary to Mfn2, Mfn1 binds PE and PC but not PS [39].

On the other hand, it is relevant to pay attention to other phospholipid transport such as PA. Since PA is the precursor of cardiolipin, PA transference between the ER and the mitochondria is essential to ensure enough levels of this phospholipid for cardiolipin production. One of the members of the conserved Ups1/PRELI-like proteins in the intermembrane space, called protein UPS1, mitochondrial (Ups1), is described to regulate the accumulation of PE and cardiolipin in the mitochondria [40]. Moreover, it has been recently shown in yeast that Ups1 enables PA transfer from the outer to the inner mitochondrial membrane. However, it remains unclear how and what kind of LTPs can bind and transfer PA from the ER to the mitochondria outer membrane.

### 3.3. Phospholipid Transport between the ER-Peroxisome

Peroxisomes are small membrane organelles involved in a few aspects of energy metabolism. This organelle contains several different enzymes such as catalases, which degrade the hydrogen peroxide that peroxisomes form during oxidation reactions. Additionally, peroxisomes are involved in lipid biosynthesis, with cholesterol and dolichol being formed in this organelle as well as in the ER. In the ER-peroxisome transport process (Figure 4), ER receives lipid precursors for plasmalogen biosynthesis from peroxisomes and peroxisomes receive lipids in order to maintain their lipid composition.

### 3.4. Phospholipid transport between the ER-Golgi

MCS between these two membrane structures assemble through vesicle-associated membrane protein-associated protein B/C (VAP-B) (in the ER) interaction by its major sperm protein (MSP) domain with the two phenylalanines (FF) in an acidic tract (FFAT)-like motif of acyl-CoA binding domain-containing 5 (ACBD5) in the peroxisomes. The four alpha-helices arranged in a bowl shape (ACB domain) of ACBD5 binds long-chain fatty acyl-CoAs, delivering them to peroxisomes [41].

It has been demonstrated that patients presenting mutations in ACBD5 exhibit retinal dystrophy [42]. In fact, when ACBD5 is mutated, there is an accumulation of high levels of very long-chain fatty acids, which correlate with an impaired peroxisome β-oxidation of these fatty acids.

The Golgi complex or Golgi apparatus is a membrane-bound organelle of eukaryotic cells constituted of four or eight cisternae. The Golgi apparatus is structurally polarized, with a cis face, the closest to the ER and a trans face, which is the farthest. Both faces and the middle cisternae are biochemically different, and the enzymatic content of each segment is markedly distinct.

Several proteins, glycoproteins and glycolipids, are delivered to the Golgi apparatus in order to be modified. Regarding phospholipids, PI is transported from the ER to Golgi to maintain PI4P levels at the Golgi during associated cholesterol transport (Figure 5).

In this case, PI is delivered at MCS through phosphatidylinositol transfer protein β (PITPβ), whereas Oxysterol Binding Protein (OSBP) is responsible for cholesterol transfer. The FFAT motif of OSBP binds to VAP at the ER and its pleckstrin homology domain (PH domain) binds to PI4P at the Golgi, and finally, its ORD domain drives cholesterol export through the reciprocal transfer of PI4P.

After PI is transported and phosphorylated in the Golgi, it is delivered back to the ER through the PH domain of OSBP, where it is dephosphorylated again to form PI [43].

### 3.5. Phospholipid Transport between the ER-Autophagosome

Autophagosome formation is a key aspect of the autophagic process. It is demonstrated that the transfer of PI from the ER is needed for autophagosome formation. Phosphatidylinositol synthase (PIS) synthesizes PI from CDP-DAG. Phagophore structures are in close proximity to a subdomain of the ER greatly enriched in PIS [44].

The Unc-51 Like Autophagy Activating Kinase 1/Autophagy Related 13 (ULK1/ATG13) complex senses upstream signals in order to enhance or prevent the downstream autophagy pathway. It localizes in this enriched subdomain and afterwards, it translocates to Autophagy Related 9A (ATG9A)-positive autophagosome precursors in a phosphatidylinositol 3-phosphate (PI3P)-dependent manner. ATG9A is a scramblase that flips phospholipids between the two membrane leaflets, hence participating in the enlargement of the phagophore membrane [45]. However, LTPs that might transport PI from this enriched subdomain to ATG9A are still unknown.

## 4. Oxidized Phospholipids and Their Health Implications

Oxidized phospholipids (OxPL) are the result of glycerophospholipids lipid peroxidation as a consequence of the oxidative stress effect on them. Specifically, their molecular characteristics depend on the nature of fatty acids that form them, the oxidation conditions, and the oxidative species. For instance, unsaturated fatty acids are prone to be oxidized while saturated ones are more resistant. Since mitochondria are one of the major reactive oxygen species (ROS) producers, relevant oxidized phospholipid sources are found in this organelle. This explains why mitochondria are associated with the principal consequences of oxidized phospholipid attacks [46,47,48].

It is demonstrated that OxPLs have a significant impact on cell activity, modulating gene expression, and inducing cellular stress and apoptosis. Furthermore, they are presented in several relevant structures, such as low-density lipoprotein (LDL) or apoptotic cells [48].

OxPL in vivo formation may or may not be catalyzed by enzymes. Some enzymes with a relevant weight are lipoxygenase, myeloperoxidase, cyclooxygenases, cytochrome p450, and Nicotinamide-Adenine dinucleotide phosphate (NADPH) oxidase (it is a ROS production enzyme). Due to OxPLs being produced, in most cases, by non-enzymatic reactions, there are not many mechanisms to avoid their formation.

Of all OxPL detected in mammals, the majority contain choline, indicating that this glycerophospholipid is one of the most oxidized. A growing amount of evidence indicates that OxPL is not equally distributed throughout the body. For example, large amounts of PC have been identified in fibrous plaques associated with atherosclerosis, as is PE in the retina [46,49].

### 4.1. Oxidized Phospholipids and Atherosclerosis

It is known that mitochondria produce significant amounts of ROS, which provokes lipid peroxidation, generating important quantities of OxPL. These phospholipids have an important role in atherosclerosis pathogenesis [50,51].

OxLPs have a key function in the clinical course of chronic diseases, such as atherosclerosis or NASH [52,53]. In atherosclerosis, the oxidation of PC (which is focused on fibrous plaques) is a central event due to its pro-inflammatory and pro-atherogenic effects. Seemingly, its important role is due to be the major component of cell membranes, LDL, and lipid droplets [49].

The presence of oxidized PC leads to monocyte recruitment by endothelial cells. This recruitment is induced by the connecting segment 1 (CS-1) fibronectin and α4β1 integrin. Specifically, oxidized phosphatidylcholine is recognized by Toll-Like Receptors (TLRs) and scavenger receptors, present in monocytes and macrophages. Moreover, it is documented that antibodies against OxPLs are present in patients with atherosclerosis diabetes or hypertension, indicating their relevance in disease development [49,52,53,54,55].

In several studies, it has also been demonstrated that high levels of OxLPs are present in patients that suffer coronary pathologies, being correlated with their disease stages [56,57]. The best correlation is with oxidized PC on apolipoprotein B-100 (apoB100). These levels are measured using an anti-oxidized phospholipid E06 antibody, which recognizes specifically the oxidized PC, but not its unoxidized state. Furthermore, the oxidized PC levels on apoB100 predict the possibility of suffering symptomatic coronary events, indicating the good prognostic value of the levels of these phospholipids [52,58].

Additionally, some aspects of the early stages of atherosclerosis development are linked to endothelial dysfunction and impaired vasodilatation. In particular, in vitro studies of human umbilical vein endothelial cells show less nitric oxide production as well as lower levels of proliferation, migration, and tube formation due to a proinflammatory lipid called 1-palmitoyl-2-(5-oxovaleroyl)-sn-glycero-3-phosphocholine [59]. This phospholipid is the oxidation product of oxidized low-density lipoprotein, which is known to activate p38 mitogen-activated protein kinase, c-jun N-terminal kinase, and caspase-3 pathway, leading to apoptosis in arterial smooth muscle cells and macrophages. Moreover, OxPL can interact with CD36 receptor present in macrophages, resulting in foam cell formation, one of the main steps in atherosclerosis development [60,61].

### 4.2. Oxidized Phospholipids, Mitochondria, and NASH

Although the implications of oxidized phospholipids in atherosclerosis development are evident, the same does not occur with NASH. However, recent studies describe the increment of these lipids in the liver and blood of murine NASH models [53,62,63,64,65].

OxPL in NASH pathogenesis is strongly associated with mitochondria. It is recently demonstrated that when targeting OxPL, an improved mitochondrial function in adipose tissue and liver is obtained. It is clear that mitochondria are one of the major ROS producers, and hence, oxidized phospholipids, as it promotes lipid peroxidation [52,53].

Additionally, there is a positive feedback loop of ROS production. Mitochondrial antioxidant manganese dismutase (MnSOD) covalently modifies OxPL, altering its function. This enzyme neutralizes free radicals generated during mitochondrial respiration. Moreover, this deregulation leads to alterations in mitochondrial membrane potential and increases ROS production [53,62,63,64,65,66].

When OxPLs are attacked by specific antibodies, the main NASH implications, such as fibrosis, inflammation, cell death, steatosis, and the progression to hepatocellular carcinoma (HCC), ameliorate. Specifically, the OxPLs blocking decreases hepatic inflammation, reducing the macrophages recruitment, inflammatory-associated gene expression, and circulating levels of pro-inflammatory cytokines [52].

These features provide evidence of OxPLs’ central role in the development of murine NASH implications. In humans, OxPLs high levels in blood and liver were also strongly related to NASH. Importantly, they are intimately linked with progression from steatosis to NASH [53,66].

These oxidized phospholipids are recognized by the TLRs and scavenger receptors located in immune cell surface as occurs in atherosclerosis. These interactions cause the activation of nuclear factor kappa-light-chain-enhancer of activated B cells (NF-κB), which provokes the proinflammatory cytokine production and increases macrophage proliferation and activation [52,53,62,63,67,68,69,70].

For these reasons, the tracing of oxidized phospholipids in peripheral circulation may be an adequate method that can help in the distinction of NASH and its progression to non-alcoholic fatty liver disease (NAFLD) and hepatic cancer with a high-fat diet or ageing. Blocking these oxidized phospholipids, through antibodies such as E06, can also be a good strategy to ameliorate these common liver pathologies [52,53,62,63,67,68,69,70].

Moreover, when OxPL is neutralized, an increase in mitochondria biogenesis is also observed, indicating the tight relationship between these two items. The OxPL blocking produces an increase in the nicotinamide adenine dinucleotide/nicotinamide adenine dinucleotide phosphate (NAD/NADH) ratio, which finally provokes major mitochondrial biogenesis through the nicotinamide adenine dinucleotide/silent mating type information regulation 2 homolog 1/peroxisome proliferator-activated receptor gamma coactivator 1-alpha (NAD/SIRT1/PGC-alfa) axis [52,53].

## 5. Deficient Phospholipid Transport and Their Consequences

### 5.1. Mfn2

Classically, Mfn2 has been considered a mitochondrial GTPase protein that only intervened in the fusion processes of the outer mitochondrial membrane (OMM). Nevertheless, in recent years, it has been demonstrated that Mfn2 has many other implications, such as regulation of the interactions between ER and mitochondria, autophagosome formation, insulin signaling, energy homeostasis, and phospholipid transport [39,71,72,73].

Hernandez-Alvarez, M.I., et al. for the first time associated Mfn2 deficiency with the development of liver disease (Figure 6). Specifically, the ablation of Mfn2 in mouse livers causes inflammation, triglyceride accumulation, fibrosis, and in the last instance, liver cancer. In addition, a reduction in the levels of Mfn2 has been observed in hepatic biopsies from patients with NASH and the levels of this protein were also lower in mouse models of steatosis or NASH. Moreover, its re-expression in a NASH mouse model ameliorated the disease.

All these conditions exposed so far are due to a reduction in Mfn2 hepatic levels, leading to a poor PS transfer and phospholipid synthesis, which causes ER stress, a NASH-like phenotype, and liver cancer.

Furthermore, Mfn2 deficiency alters phospholipid metabolism by inhibiting PS synthesis as a consequence of less PSS1 and PSS2 compensatory expression. The lack (or reduction) of Mfn2 also generates MAMs remodeling (altering the phospholipid composition in ER-mitochondrial contact sites), leading to a triglyceride accumulation, insulin resistance, and deregulation of phospholipid synthesis [39].

The importance of phospholipid deregulation during NASH development is supported by a study where NAFLD was shown to alter phospholipid zonation, changing their distribution based on the state and severity of liver disease [74].

### 5.2. PITP

From all phospholipids, PI, particularly PI(4,5)P_2_ and IP_3,_ are linked to cancer [75,76]. Since PI(4,5)P_2_ is quickly hydrolyzed by PLC in order to produce second messengers, a PI impaired transport could cause defects in cellular processes regulated by PLC signaling. Moreover, pathways related to cell proliferation and growth, such as protein kinase B (PKB), also known as the AKT pathway, are upregulated in several human cancers [77]. In fact, a considerable number of studies associate alterations in the expression of PITPα in mouse fibroblasts, leading to cell proliferation, or Nir2, which has been shown to enhance breast cancer metastasis [78,79].

### 5.3. ORP5/8

Regarding ORP8 phospholipid transfer protein, it has been demonstrated to be significantly downregulated in primary HCC cells and cell lines, protecting liver cancer cells from ORP8-mediated apoptosis [80]. Nevertheless, there is not much evidence related to diseases caused by alterations in ORP5/8-phospholipid transport. It is intriguing to contemplate its possible impact as this protein contributes to the tight ER-PM phospholipid transport and regulation. In parallel with this speculation, it has been recently proven that depletion of ORP5/8 induced a decrease in PS levels from PM, leading to mislocalization of Kirsten rat sarcoma virus (KRas), an oncogene from the PM whose activity relies on PS levels [81]. Although these proteins are relevant in keeping with lipid equilibrium and signaling responses, further analyses are needed to elucidate their possible lipid-related diseases.

### 5.4. Cardiolipin

Cardiolipin directly interacts with multiple mitochondrial proteins, including complexes of the respiratory chain and ATP production proteins. Moreover, CL is an important target of oxidative damage due to the high unsaturated fatty composition of this mitochondrial lipid. Therefore, changes in cardiolipin will have a negative impact on mitochondrial function [20,82]. For instance, the Barth syndrome is an X-linked genetic disorder directly associated with cardiolipin alterations. This pathology caused severe clinical manifestations (enlarged heart, low blood cell count, and muscle weakness, among others), and it is provoked by mutations in a gene encoding tafazzin, a transacylase that promotes CL fatty acid chain remodeling. Mutation in this gene decreases cardiolipin levels, altering the inner mitochondrial membrane cristae and both mitochondrial morphology and function [20,83,84,85]. On the other hand, it has been observed that there is an altered content of CL in diabetic animal models. Specifically, there is a decrease in CL levels and alterations in its fatty acid chain remodeling, increasing the presence of long-chain unsaturated fatty acids. Moreover, acyl-CoA:lysocardiolipin acyltransferase-1 (ALCAT1), an essential acyltransferase in CL remodeling, is upregulated during oxidative stress states. This leads to the presence of long-chain unsaturated fatty acids CL, which is more susceptible to peroxidation, mitochondrial dysfunction, and insulin resistance. Cardiolipin also interacts with key proteins related to mitochondrial dynamics, such as Mitofusins, Dynamin-related Protein 1 (Drp1), and OPA1. In parallel with defects of CL in diabetic animals, mitochondrial dynamics proteins have also been linked to insulin resistance. Particularly, mitochondrial fission is increased in diabetic and obese models [20,86,87,88,89].

## 6. Conclusions

It is clear that both phospholipid transport and distribution are crucial in maintaining membrane lipid composition. Additionally, they participate in signaling pathways. Their aberrant transport could lead to impairment of cellular processes, causing several diseases already commented on in this review, such as NASH, insulin resistance, glucose intolerance, or the Barth syndrome, and some others still unknown. Within the phospholipid’s delivery process, a large number of molecules are harmonically coordinated. In parallel with this, phospholipid transfer proteins are established as essential in phospholipid handling, especially in the small hydrophobic gaps formed between the outer membrane of two organelles or MCS. This specific area allows faster phospholipid circulation, which is extremely important when it comes to signaling pathways that control cell growth or apoptosis. However, the impaired phospholipid delivery is not the only involved process, which causes phospholipid-related diseases. Depending on the oxidative conditions, oxidized phospholipids can be formed. These phospholipids are different in their fatty acid composition, and they modulate relevant cellular processes, such as cellular stress or gene expression.

Although a considerable number of researchers are recently focusing on elucidating the underlying mechanisms, such as functions of lipid transfer proteins and molecular interactions within the phospholipid delivery process, there are still many pathways to investigate. An example would be the continuous discovery of new proteins capable of transporting lipids, including Mfn2 and ORP5/8 in MAMs.

As far as oxidized phospholipids are concerned, targeting them can be a good strategy to ameliorate diseases related with their actions. For instance, it is demonstrated that blocking OxPL produces a reduction of NASH severity through reducing fibrosis, inflammation, and cell death and increasing mitochondrial biogenesis.

Therefore, as LTPs and OxPL are directly associated with pathological processes and targeting, inhibiting OxPL or restoring LTP functions would help us to develop new approaches to treat relevant diseases.

## Figures and Tables

**Figure 1 biomedicines-10-01201-f001:**
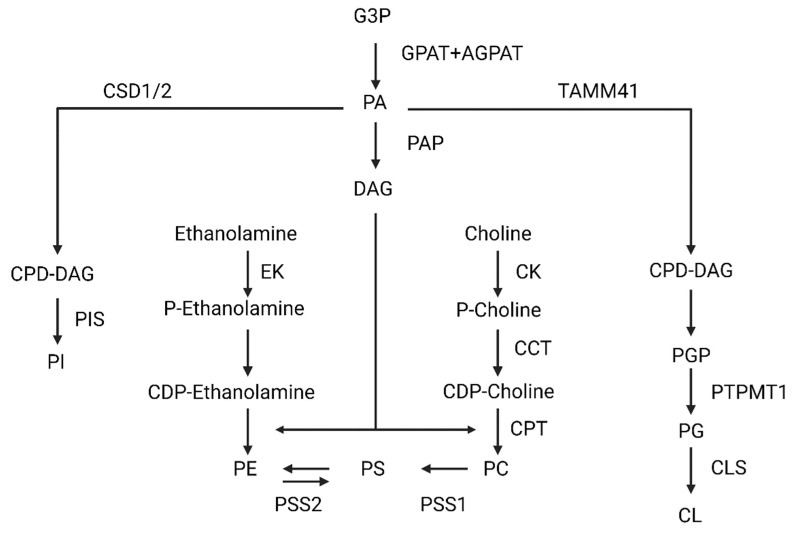
Scheme of phosphatidylcholine (PC), phosphatidylserine (PS), phosphatidylethanolamine (PE), cardiolipin (CL), and phosphatidylinositol (PI). Abbreviations: glycerol-3-Phosphate (G3P); glycerol-3-phosphate acyltransferase (GPAT); 1-acylglycerol-3-phosphate-O-acyltransferase (AGPAT); phosphatidic acid (PA); phosphatidic acid phosphatase (PAP); diacylglycerol (DAG); ethanolamine kinase (EK); phosphatidylserine synthase 2 (PSS2); phosphatidylserine synthase 1 (PSS1); CDP-choline:1,2-diacylglycerol cholinephosphotransferase (CTP); choline-phosphate cytidylyltransferase (CCT); choline kinase (CK); Cytidine diphosphate diacylglycerol synthetases 1/2 (CSD1/2); cytidine diphosphate diacylglycerol (CDP-DAG); (PIS); Mitochondrial translocator assembly and maintenance protein 41 (TAMM41); phosphatidylglycerophosphate (PGP); Protein Tyrosine Phosphatase Mitochondrial 1 (PTPMT1); phosphatidylglycerol (PG); cardiolipin synthase (CLS).

**Figure 2 biomedicines-10-01201-f002:**
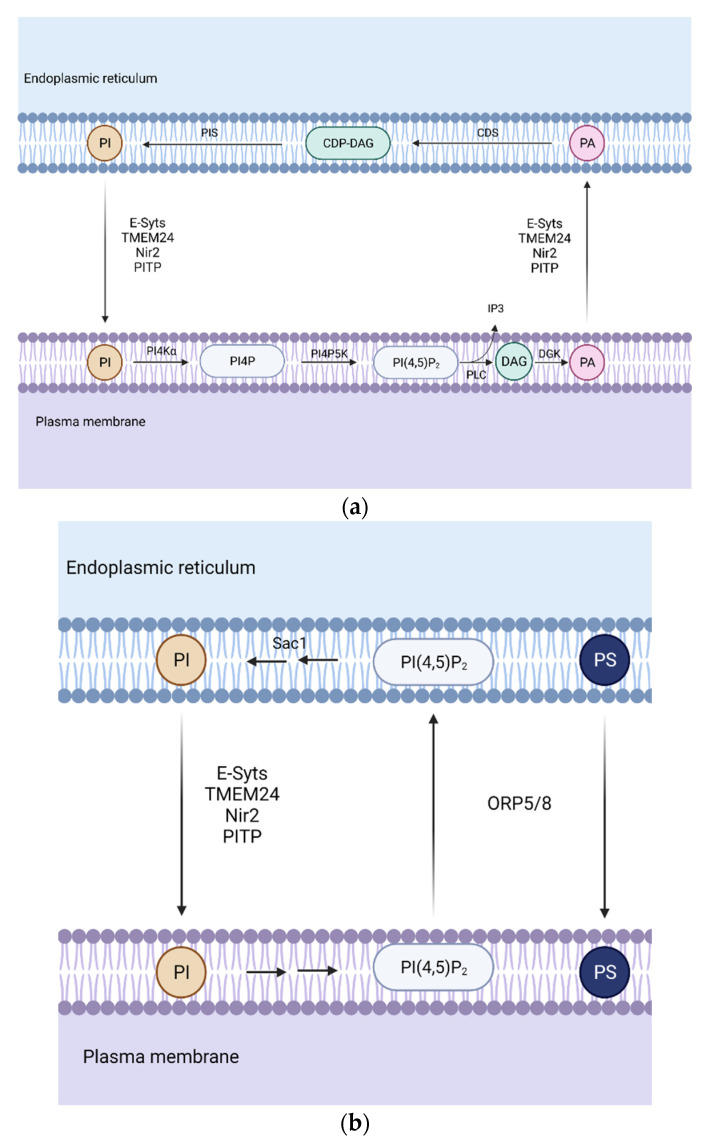
Representation of the phospholipid transport during phospholipase C (PLC) signaling. (**a**) Association between the endoplasmic reticulum (ER) and the plasma membrane (PM) is required in order to ensure high levels of phosphatidylinositol 4,5-biphosphate PI(4,5)P_2_ which can drop quickly during PLC signaling. (**b**) Phosphatidylserine (PS) transport and PI(4,5)P_2_ and phosphatidylinositol 4-phosphate PI4P removal. Oxysterol-binding protein (OSBP)-related protein 5/8 (ORP5/8) removes PI(4,5)P_2_ and PI4P from the PM and transfers PS from the ER to PM in exchange, creating a gradient necessary for phosphatidylinositol (PI) transport. Abbreviations: PI4Kα, phosphatidylinositol 4-kinase; PI4P5K, phosphatidylinositol 4-phosphate 5-kinase; DGK, diacylglycerol kinase; PIS, phosphatidylinositol synthase; CDS, CDP-diacylglycerol synthase; synaptotagmins (E-Syts); the Phospholipid Transfer Protein C2CD2L (also known as TEM24 or C2CD2L); membrane-associated phosphatidylinositol transfer protein 1 (Nir2); phosphatidylinositol transfer protein (PITP).

**Figure 3 biomedicines-10-01201-f003:**
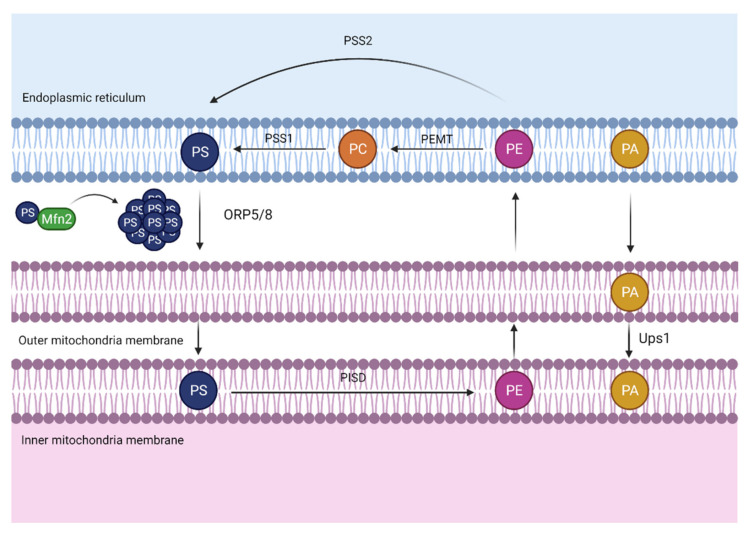
Mitochondria-associated membranes (MAMs). The close apposition between the endoplasmic reticulum (ER) and the mitochondria allows the transport of phosphatidylserine (PS) from the ER to mitochondria, where it can be converted into phosphatidylethanolamine (PE) and transported back to the ER. The PS transport is facilitated due to a mitochondrial outer membrane nucleotide guanosine triphosphate phosphatase (GTPase), mitofusin 2 (Mfn2), which elaborates rigid domains enriched in PS, hence, facilitating PS transport through oxysterol binding protein-like 5/8 (ORP5/8). On the other hand, protein UPS1 mitochondrial (Ups1) transfers phosphatidic acid (PA) from the outer to the inner mitochondrial membrane. Abbreviations: PSS1, phosphatidylserine synthase 1; PSS2, phosphatidylserine synthase 2; PEMT, phosphatidylethanolamine *N*-methyltransferase; PISD, phosphatidylserine decarboxylase proenzyme; PC, phosphatidylcholine.

**Figure 4 biomedicines-10-01201-f004:**
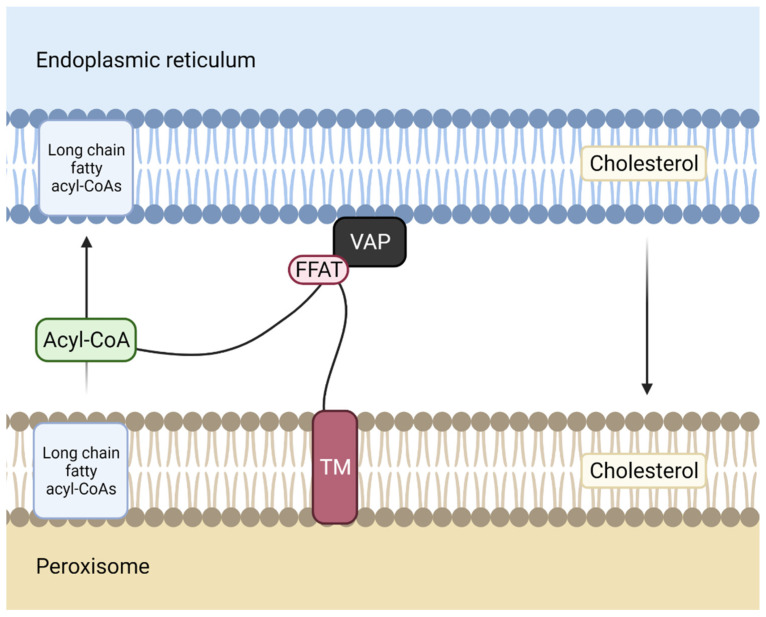
Membrane contact site between the ER and the peroxisome. Acyl-CoA binding domain-containing 5 ACBD5, a protein anchored to peroxisome membrane through its transmembrane domain (TM), interacts with vesicle-associated membrane protein-associated protein (VAP), an endoplasmic reticulum (ER) membrane protein, via its two phenylalanines (FF) in an acidic tract (FFAT)- like motif. The Acyl-CoA domain of ACBD5 transports long-chain fatty acyl-CoAs from the ER to the peroxisome.

**Figure 5 biomedicines-10-01201-f005:**
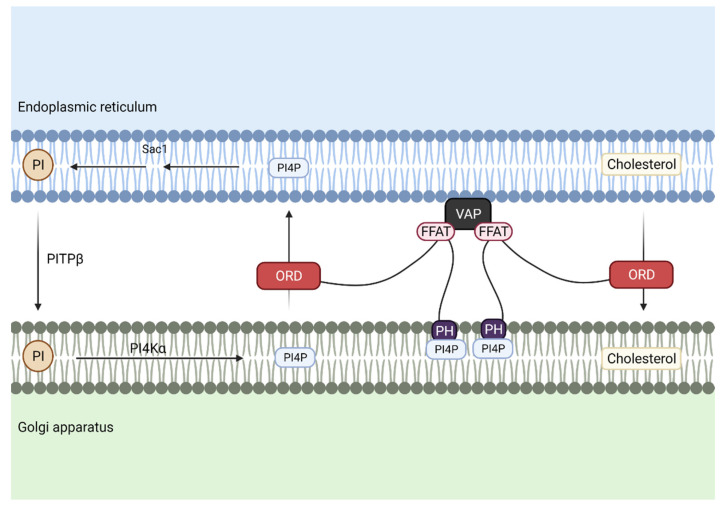
Cholesterol and phosphatidylinositol 4-phosphate (PI4P) are transported between the endoplasmic reticulum (ER) and the Golgi apparatus. To maintain PI4P levels associated with cholesterol transport, phosphatidylinositol (PI) is transferred through phosphatidylinositol transport protein β (PITPβ), while the OxySterol Binding Protein (OSBP)-Related Domain (ORD motif) of Oxysterol Binding Protein (OSBP) transports cholesterol from the ER to the Golgi membrane and reciprocal transport of PI4P from Golgi membrane to the ER. Association of these two membrane structures is achieved by the interaction of the two phenylalanines (FF) in an acidic tract (FFAT)-like motif of OSBP with vesicle-associated membrane protein-associated protein (VAP) in the ER, and pleckstrin homology domain (PH domain) from OSBP, which binds to PI4P at the Golgi.

**Figure 6 biomedicines-10-01201-f006:**
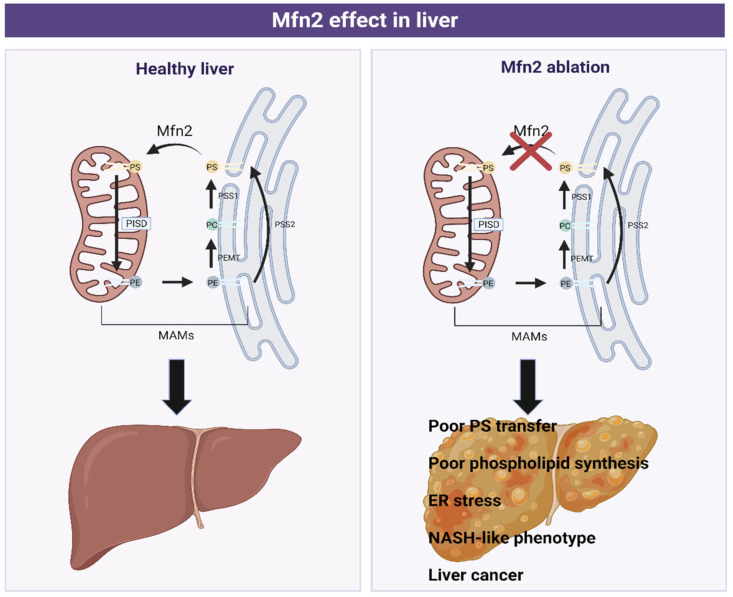
Graphical scheme of phosphatidylserine (PS), phosphatidylethanolamine (PE), and phosphatidylcholine (PC) synthesis in Mitochondria-Associated Membranes (MAMs) in healthy (left) and pathological (right) liver conditions. Phosphatidylcholine (PC) in the ER is transformed to phosphatidylserine (PS) (catalyzed by the phosphatidylserine synthase 1 (PSS1)), which is transported to mitochondria by Mitofusin 2 (Mfn2). Here, it is converted to phosphatidylethanolamine (PE) by phosphatidylserine decarboxylase proenzyme (PISD), a protein located in the mitochondrial membrane. PE is then transferred to ER, where is converted to PC or PS, depending on the enzyme implicated, phosphatidylethanolamine *N*-methyltransferase (PEMT), or phosphatidylserine synthase 2 (PSS2), respectively.

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
