# Peer review of "Phospholipid Membrane Transport and Associated Diseases"

_biomedicines, 2022, doi:10.3390/biomedicines10051201_

Round 1

Reviewer 1 Report

This review article introduces several main classes of phospholipids and the molecular mechanisms of their syntheses, transport and functions, as well as their roles in NAFLD and NASH. This article brings basic and novel knowledge of phospholipids to readers. Some points can be improved.

  1. Figures need to be referred in the text.
  2. Line 176, the word space between PIPT and N-terminal is too large.
  3. There are many abbreviations in this article but some of them without explanation. Please check carefully and add the full terms or explanation. 
  4. The title "Phospholipid function in membrane transport and associated diseases" is not so appropriate. The reviewer's understanding after reading this review is that transport of phospholipids between two different locations is rely on lipid transport proteins (LTPs). Thus, phospholipids have no function in membrane transport but LTPs have the function for membrane transport. In most of the parts, this review is explaining how phospholipids are transported by the related proteins. 

Author Response

We thank the referees for their kind review of this manuscript. We have made the suggested changes and believe it has improved the document. We list the specific changes we have made as follows:

Title change -> line 2

Abbreviation correction -> lines 86, 87, 97, 98, 102, 103, 115, 116, 117, 133, 134, 137, 138, 147, 149, 150, 154, 155, 156 174, 175, 180, 181, 182, 183, 189, 190, 197, 204, 205, 207, 213, 214, 215, 216, 217, 218, 219, 220, 221, 222, 223, 224, 228, 268, 275, 276, 278-281, 292, 293, 294, 295, 296, 303, 304, 305, 306, 307, 318, 319, 320, 321, 326-334, 340, 341, 343, 344, 354, 363, 430, 440-444, 475, 476, 478, 486, 487, 493, 499, 517,

OxPL role in atherosclerosis and endothelial dysfunction -> lines 394-404

Information about promising directions for future research that would help to broaden the understanding of the role of phospholipids in some diseases -> lines 542-553

Clarification between NAFLD and NASH-> lines 434, 435

Figures referred in the text -> Lines 85, 171, 205, 239, 290, 318, 453

Reviewer 2 Report

The review article is of clinical and research interest. The review contains well-structured information about phospholipids. Minor comments:

  1. It is recommended to check again all abbreviations (for example, in line 18 "membrane contact sites" is abbreviated as "MCS" and in line 147 as "MSC"; in line 88 there is probably a typo in "acylglyerophosphate acyltransferase (AGPAT)", etc.
  2. It is recommended that all abbreviations be deciphered in the figure descriptions;
  3. The role of phospholipids in atherosclerosis could be expanded by describing their involvement in endothelial dysfunction;
  4. Based on lines 403-405, it can be understood that NAFLD and NASH are different diseases, although NASH is considered to be a form of NAFLD. It is recommended that this sentence be corrected.
  5. It is recommended to add information about promising directions for future research that would help to broaden the understanding of the role of phospholipids in some diseases.

Author Response

(The authors gave the same response as above.)
